# Identifying essential genes in genome-scale metabolic models of consensus molecular subtypes of colorectal cancer

Chao-Ting Cheng[1], Jin-Mei Lai[2], Peter Mu-Hsin Chang[3,4], Yi-Ren Hong[5], Chi-Ying F. Huang[6,7], Feng-Sheng Wang[1]*

1 Department of Chemical Engineering, National Chung Cheng University, Chiayi, Taiwan, 2 Department of Life Science, College of Science and Engineering, Fu Jen Catholic University, New Taipei City, Taiwan, 3 Department of Oncology, Taipei Veterans General Hospital, Taipei, Taiwan, 4 Faculty of Medicine, National Yang Ming Chiao Tung University, Taipei, Taiwan, 5 Department of Biochemistry and Graduate Institute of Medicine, Kaohsiung Medical University, Kaohsiung City, Taiwan, 6 Institute of Biopharmaceutical Sciences, National Yang Ming Chiao Tung University, Taipei, Taiwan, 7 Department of Biotechnology and Laboratory Science in Medicine, National Yang Ming Chiao Tung University, Taipei, Taiwan

* chmfsw@ccu.edu.tw

**Data Availability Statement:** The source programs of anticancer target discovery platform and the tissue-specific CMS genome-scale metabolic models are coded by the General Algebraic

## Abstract

Identifying essential targets in the genome-scale metabolic networks of cancer cells is a time-consuming process. The present study proposed a fuzzy hierarchical optimization framework for identifying essential genes, metabolites and reactions. On the basis of four objectives, the present study developed a framework for identifying essential targets that lead to cancer cell death and evaluating metabolic flux perturbations in normal cells that have been caused by cancer treatment. Through fuzzy set theory, a multiobjective optimization problem was converted into a trilevel maximizing decision-making (MDM) problem. We applied nested hybrid differential evolution to solve the trilevel MDM problem to identify essential targets in genome-scale metabolic models for five consensus molecular subtypes (CMSs) of colorectal cancer. We used various media to identify essential targets for each CMS and discovered that most targets affected all five CMSs and that some genes were CMS-specific. We obtained experimental data on the lethality of cancer cell lines from the DepMap database to validate the identified essential genes. The results reveal that most of the identified essential genes were compatible with the colorectal cancer cell lines obtained from DepMap and that these genes, with the exception of EBP, LSS, and SLC7A6, could generate a high level of cell death when knocked out. The identified essential genes were mostly involved in cholesterol biosynthesis, nucleotide metabolisms, and the glycerophospholipid biosynthetic pathway. The genes involved in the cholesterol biosynthetic pathway were also revealed to be determinable, if a cholesterol uptake reaction was not induced when the cells were in the culture medium. However, the genes involved in the cholesterol biosynthetic pathway became non-essential if such a reaction was induced. Furthermore, the essential gene CRLS1 was revealed as a medium-independent target for all CMSs.

Modeling System (GAMS, https://www.gams.com/), and are available in http://doi.org/10.5281/zenodo.7136561. Supplementary tables have been provided in a Microsoft Excel format along with this article.

**Funding:** MOST111-2320-B-194-003 and MOST111-2221-E-194-004 to FSW MOST111-2320-B-030-008 to JML MOST111-2320-B-075-009 to PMHC MOST111-2320-B-037-028;KMU-DK (A)111006 to YRH MOST111-2320-B-A49-036 to CYFH The funders had no role in study design, data collection and analysis, decision to publish, or preparation of the manuscript.

**Competing interests:** The authors have declared that no competing interests exist.

## Introduction

Colorectal cancer (CRC) is a major health burden worldwide, and among cancers, it ranks third and second in terms of its incidence and mortality, respectively [1], indicating the global need for improved prognoses and treatment strategies. More than 1.9 million new CRC (including anal cancer) cases and 935000 deaths were estimated to have occurred in 2020; indicating CRC was responsible for approximately 1 in 10 cancer cases and deaths [1]. The classification of CRC plays a pivotal role in predicting a patient's prognosis and determining treatment strategies. The tumor, node, and metastasis classification system is commonly used to determine the progression of CRC. However, in-depth characterization is required to improve the assessment of treatment strategies and prognoses. The consensus molecular subtype (CMS) system is an RNA expression-based classification system for CRC. It was developed using 18 CRC data sets and, as of 2015, contains 4151 CRC samples [2]. CRC can be classified into four subtypes, and each subtype exhibits distinct molecular and biological characteristics and pathological and genetic signatures. The development of molecular subtype-based therapies has provided a new framework for implementing preferred and precise medical treatments. Several studies have used CMS classification to predict the prognosis of patients with CRC and to determine treatment strategies [3–9].

Tissue-specific genome-scale metabolic models (GSMMs) are frequently applied to identify anticancer targets and obtain insight into the metabolic bases of physiological and pathological processes [10–30]. The Cancer Genome Atlas (TCGA) [31] and Human Protein Atlas (HPA) [32] have been incorporated into models of human metabolic networks, such as Recon X [32–37] and Human-GEM [24, 38], to reconstruct tissue-specific GSMMs to investigate metabolic processes. However, neither TCGA and HPA has been used to reconstruct CMS-based tissue-specific GSMMs. Furthermore, a literature review revealed that no study has identified targets for CRC by integrating CMS classification with GSMMs. The present study incorporated CMS classification data for CRC samples from TCGA [2, 31] into Recon3D to reconstruct the CMS-specific GSMMs of CRC. We applied these CMSs to identify essential genes, metabolites, and reactions and used a fuzzy decision-making method to evaluate the cancer cell mortality, healthy cell viability, and metabolic perturbation effects resulting from the blockage of the corresponding fluxes.

The present study proposed a fuzzy hierarchical optimization framework for identifying essential genes for the treatment of each CMS of CRC. The framework of the present study is an extension of the identifying anticancer targets framework [25, 26]; it provides RNA-sequencing (RNA-seq) expression in inner optimization problems and yields uniform flux patterns for treated and perturbed cells. Traditional cell culture media (i.e., Dulbecco's Modified Eagle Medium [DMEM] and Ham's medium), were prepared to ensure continuous cancer cell proliferation in vitro. However, the composition of these media may not fulfill the nutritional requirements of tumor cells. Studies have demonstrated that the use of specific medium components can yield cell culture results that evidence alternations in tumor metabolism [39–41]. Therefore, the present study used various media that induced uptake reactions to investigate the influence of nutritional components on tumor cell growth and identify essential targets for the CMSs of CRC.

## Materials and methods

Essential genes, when deleted or knocked out, lead to cancer cell death or severe proliferation defects. However, healthy cells are unlikely to be affected by deletion of such genes. The present study developed an anticancer target discovery (ACTD) platform for identifying essential genes, the deletion of which causes cancer cell death while allowing healthy cells to survive

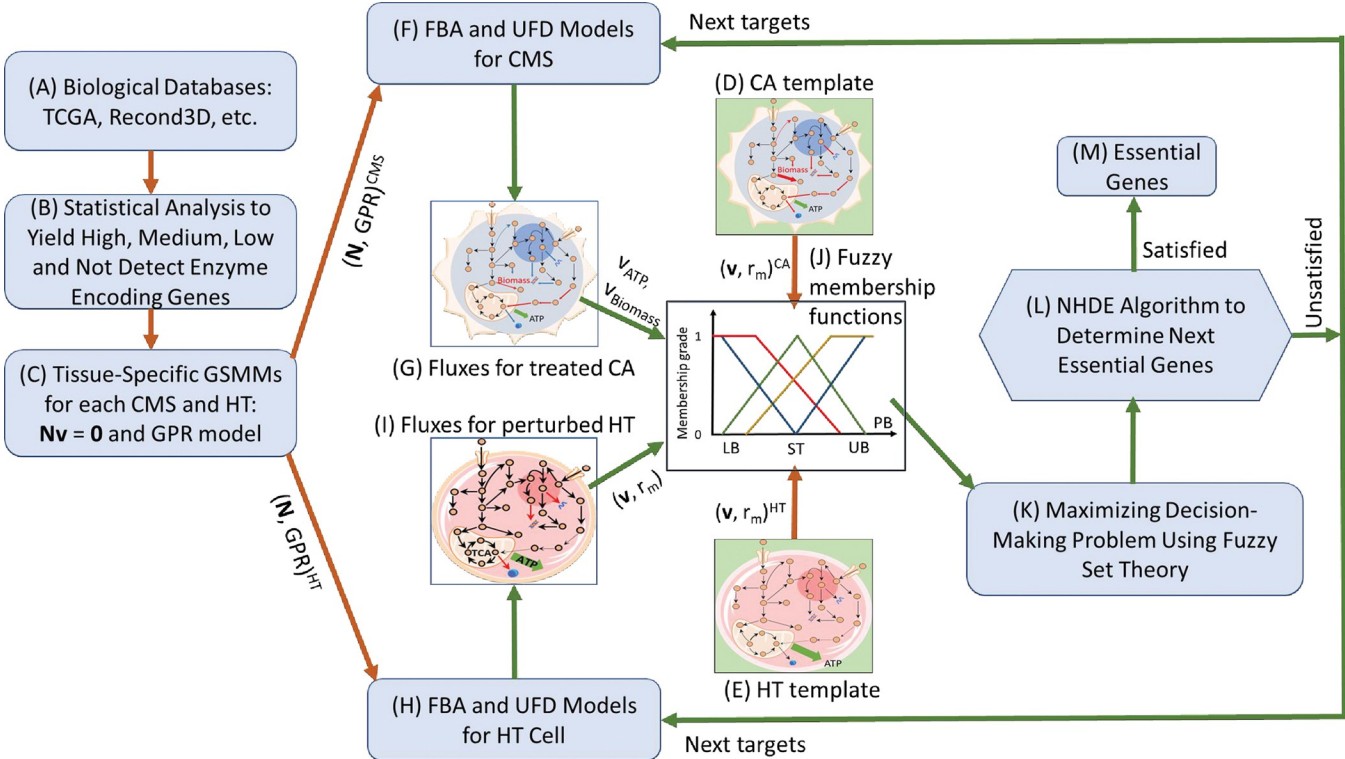

**Fig 1. Framework for identifying essential anticancer genes.** (A) Biological data, such as those on RNA-sequencing (RNA-seq) expression levels for cancerous (CA) cells and healthy (HT) cells, and human genome-scale metabolic networks, are retrieved. (B) Statistical analysis of the accessed RNA-seq expression data to performed to yield high, medium, low, and undetected enzyme encoding genes. (C) Tissue-specific genome-scale metabolic models and gene-protein-reaction models are reconstructed for each consensus molecular subtype (CMS) and HT models, respectively. (D) Flux distribution patterns for each CMS are derived from clinical data (if available); otherwise a CA template is computed through flux balance analysis (FBA) and uniform flux distribution (UFD) problems without consideration of dysregulated restriction. (E) The flux distribution patterns of HT cells can be derived from clinical data (if available); otherwise, an HT template is constructed using FBA and UFD problems without consideration of dysregulated restriction. (F) A set of candidate genes is identified using the nested hybrid differential evolution (NHDE) algorithm and used in the FBA and UFD models for each CMS to compute the treatment fluxes. (G) Flux distribution and metabolite flow rates for each candidate treatment are obtained. (H) Identical genes are used in the FBA and UFD problems for the HT cells during treatment. (I) The flux distribution and metabolite flow rates of perturbed HT cells for each candidate treatment are obtained. (J) Fuzzy membership functions for each fuzzy objective are defined to enable the evaluation of a decision criterion. (K) A multiobjective optimization problem is converted into a maximizing decision-making problem, which is solved using the NHDE algorithm. (L) Optimal essential genes are identified on the basis of the decision criterion; otherwise, Steps (F) to (L) are repeated for the subsequent set of candidate genes generated by the NHDE algorithm.

with a few side effects. Fig 1 presents the workflow of the ACTD platform, which was designed to mimic a wet-lab process for identifying essential genes.

## Tissue-specific genome-scale metabolic models

The present study integrated CMS classification data [2] for CRC samples from TCGA [31] with the human metabolic network Recon3D [37] to reconstruct CMS-based tissue-specific GSMMs for CRC and healthy cell (Fig 1A–1C). The RNA-seq expression data of 51 healthy colorectal samples (41 colon and 10 rectum samples) were downloaded from TCGA. A total 478 colonic adenocarcinoma and 166 rectal adenocarcinoma samples were obtained.

We retrieved the CMS classification criteria for the CRC samples of TCGA from the supplementary information in [2]. Table 1 presents the CMS classification of the CRC samples of TCGA. Four subtypes with distinct molecular and biological characteristics and pathological

**Table 1. Classification of colorectal cancer samples accessed from The Cancer Genome Atlas by consensus molecular subtype (CMS).** CMS classification is based on literature [2]. Samples that could not be classified under one of the four subtypes were classified as CMS5.

|            | CMS1 | CMS2 | CMS3 | CMS4 | CMS5 |
|------------|------|------|------|------|------|
| No. Samples | 83   | 226  | 72   | 151  | 112  |

and genetic signatures were identified, namely CMS1 (microsatellite instability immune), CMS2 (canonical), CMS3 (metabolic), and CMS4 (mesenchymal). The samples that could not be classified into one of the four aforementioned subtypes were categorized as CMS5 (unknown).

Tissue-specific GSMMs can be used to understand the metabolic behaviors of various physiological and pathological processes and to investigate specific cell phenotypes. Available automated reconstruction algorithms can be broadly categorized as employing flux-dependent or pruning methods [42]. Flux-dependent methods [10, 43–45] identify an optimal genome-scale metabolic network through general reconstruction and provide the maximum number of high-confidence reactions (i.e., reactions whose presence is supported by substantial experimental data). By contrast, for pruning methods [21, 46, 47], a core set of reactions is obtained from literature reviews or experimental data, and the remaining reactions in the general reconstruction are removed as functionality is maintained in the core set. The aim for both types of algorithm is to ensure the final tissue-specific reconstruction is as concise as possible. The cost optimization reaction dependency assessment (CORDA) algorithm [48] is based on the dependency assessment method, which involves identifying the dependency of desirable reactions (i.e., reactions with high experimental evidence) on undesirable reactions (i.e. reactions with no experimental evidence). We used the CORDA algorithm to reconstructed CMS-specific GSMMs and healthy model by using their corresponding RNA-seq expression data (Table 1).

Each CMS and healthy samples was analyzed using the following statistical methods to determine desirability of reactions that could be used in the CORDA algorithm. Quantile normalization was applied to normalize the raw data from healthy and CMS samples to compute the mean, confidence interval, and coefficients of dispersion (COD) for each gene. The CODs were then used to identify supportive genes and determine the differential expression of enzyme-encoding genes between each CMS and its healthy cell counterpart. The quartile method was applied to categorize the means of RNA-seq expression into four levels. Recon3D contains 2247 enzyme-encoding genes, which were classified into four levels on the basis of their participation levels (i.e., high, medium, low, and undetected). Confidence reactions were also divided into four groups (i.e., high, medium, negative, and others) on the basis of the gene-protein-reaction (GPR) association in Recon3D and their corresponding participation levels. This classification of confidence reactions was used in the CORDA algorithm [48] to reconstruct each CMS-specific and its HT GSMM.

Completing genome-scale reconstructions by using stoichiometric models for metabolites and reactions can be used to reveal mechanistic links between genotypes and phenotypes. GPR associations are typically implemented by applying Boolean rules, which enables the metabolic reactions in stoichiometric models to be linked to the gene-encoded enzymes in cells. A reaction can be catalyzed by an enzyme or isozymes. Moreover, reactions may be regulated by duplicate enzymes; that is, more than one enzyme can catalyze the same reactions. The duplicate enzymes in a GSMM can be omitted to obtain a weaker GPR association and to thereby avoid the numerous computation steps of evolutionary optimization procedures. Moreover, we use one of duplicate enzymes as a representative enzyme in the computation. Deletion of

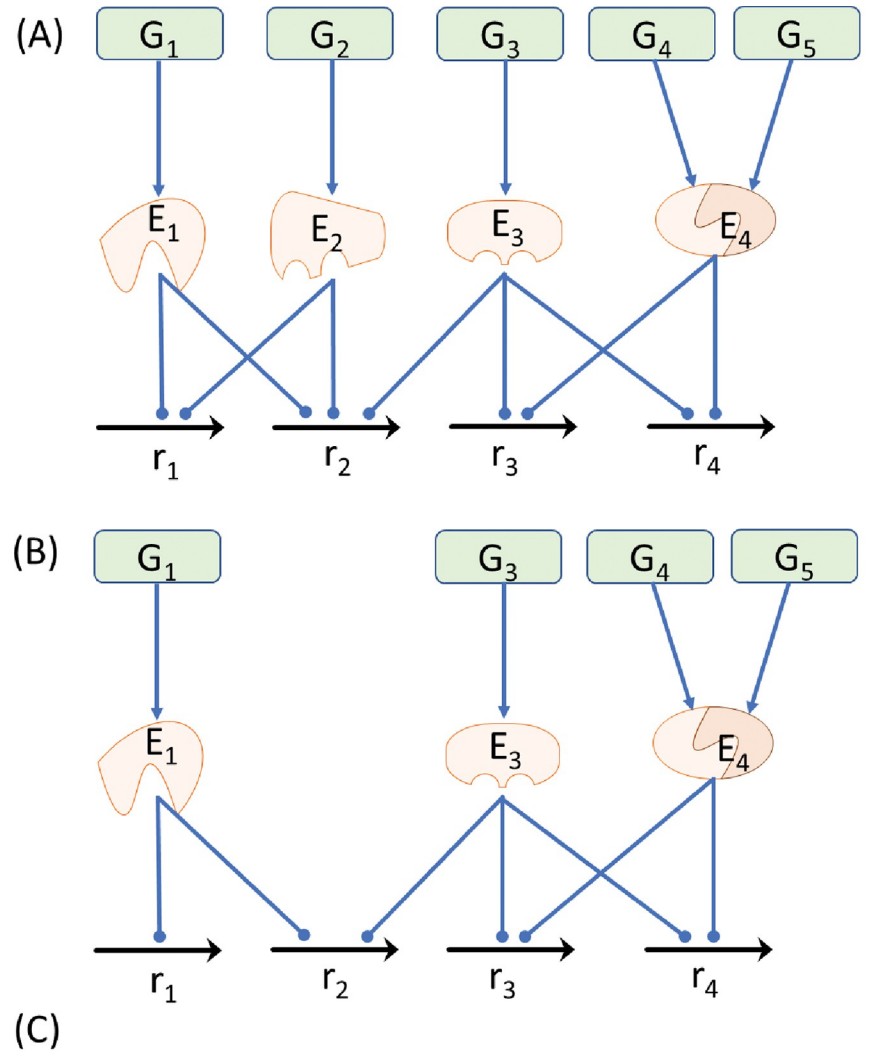

**Fig 2. Reduced gene association in a simple network.** (A) Four reactions and their gene associations. (B) Reduced model obtained by omitting duplicate genes. (C) Boolean rules for gene association. The genes $G_1$ and $G_2$ are duplicate genes that regulate the same reactions ($r_1$ and $r_2$). Therefore, one can be omitted to form a reduced gene association.

the representative enzyme in a weaker GPR association reveals that all identical duplicate enzymes are knock-out simultaneously in its original GPR association. Fig 2 depicts the process of forming a reduced gene association through duplicate gene omission in a simple network. The reaction $r_1$ is modulated by the genes $G_1$ and $G_2$ in the original GPR association of

Fig 2A. Deletion of $G_1$ is unable to block the reaction $r_1$ because the reaction is still modulated by the isozyme $G_2$ in Fig 2A. However, in the computation, we can use the reduced GPR association in Fig 2B to block the reaction $r_1$ through deletion of $G_1$ (indicated deletion of both $G_1$ and $G_2$). We developed a systems biology program that automatically builds stoichiometric and reduced GPR models in the files of the General Algebraic Modeling System (GAMS, https://www.gams.com/) for computation. The procedures were detailed in another study [25].

## Discovery of essential genes

This study presents a fuzzy hierarchical optimization ACTD framework, which was developed to mimic a wet-lab process for identifying essential genes (Fig 1D–1M; Table 2).

The ACTD platform was directly used to compute fluxes and metabolite flow rates for evaluating the fuzzy objectives. The method of the present study differs from that of the identifying anticancer targets framework, which involves logarithmic fold changes between dysregulated distribution and template levels. Evaluations using the identifying anticancer targets framework may produce numerical inaccuracies if an evaluated value is close to zero. Furthermore, in the ACTD platform, by targeting cancerous (CA) cell mortality, both the cell growth rate and adenosine triphosphate (ATP) production rate can be minimized.

The primary objective of the present study's proposed framework is to evaluate whether the growth rate of cancer cells identified for treatment (treated CA cells, denoted by TR cells) is as low as possible, which is a common requirement for discovering target problems [13, 17]. We also sought to ensure that the ATP production rate of TR cells was as low as possible. These objectives can be achieved through the fuzzy minimization of the growth rate and ATP production rate of TR cells as follows:

$$\begin{cases} \widetilde{\min_z} \ v_{biomass}^{TR} \approx 0 \\ \widetilde{\min_z} \ v_{ATP}^{TR} \approx 0 \end{cases} \tag{1}$$

An anticancer target may interfere with HT cells (perturbed HT cells, denoted as PH cells) and cause toxicity-induced tumorigenesis and harmful metabolic perturbations in HT cells. The growth rate of PH cells should be as low as possible. Perturbations in HT cells can lead to superior cell viability, which in turn maximizes the ATP production rate. The objectives can be expressed through the fuzzy minimization of the growth rate and fuzzy maximization of the

**Table 2. Optimization platform for anticancer target discovery for evaluating the performance of identified essential genes on basis of four fuzzy objectives.**

**Objectives for the outer optimization problem**
1. To evaluate the cell mortality of cancer cells following gene knockout or deletion.
2. To maximize the cell viability of perturbed healthy cells during gene knockout or deletion.
3. To evaluate the metabolic deviation of the perturbation of healthy cells as a large dissimilarity relative to that of the cancer template.
4. To evaluate the metabolic deviation of perturbation as a close similarity relative to that of the healthy template.

**subject to constraint-based models for inner optimization problems**
1. FBA and UFD problems for treating cancer cells
2. FBA and UFD problems for evaluating the perturbation of healthy cells because of treatment

ATP production rate of PH cells as follows:

$$\begin{cases} \widetilde{\min}_{\mathbf{z}} \ v_{biomass}^{PH} \approx 0 \\ \widetilde{\max}_{\mathbf{z}} \ v_{ATP}^{PH} \approx v_{ATP}^{\max} \end{cases} \tag{2}$$

The metabolic flux distributions of PH cells may be altered by cancer treatment. This side effect is evidenced by metabolic deviations, which can be verified through an evaluation of the dissimilarity of PH flux distribution relative to that of a CA template and an evaluation of similarity of PH flux distribution relative to the flux distribution of an HT counterpart. Therefore, to evaluate the grade of side effects, we defined two types of metabolic deviations for PH cells: differences in flux distributions in PH cells relative to those of a CA template and those of an HT template, respectively.

The third goal of the proposed framework was to identify the metabolic deviations of PH cells that were the most dissimilar to the CA template. This was expressed through the fuzzy dissimilarity of fluxes and metabolite flow rates of PH cells relative to those of the CA template as follows:

$$\begin{cases} \widetilde{\text{dissimilarity}}_{\mathbf{z}} \ v_j^{PH} \approx v_j^{CA} \\ \widetilde{\text{dissimilarity}}_{\mathbf{z}} \ r_m^{PH} \approx r_m^{CA} \end{cases} \tag{3}$$

The fourth goal of the proposed framework was to determine the fuzzy similarity of the fluxes and metabolite flow rates of the PH cells relative to those of the HT template, which was expressed as follows:

$$\begin{cases} \widetilde{\text{similarity}}_{\mathbf{z}} \ v_j^{PH} \approx v_j^{HT} \\ \widetilde{\text{similarity}}_{\mathbf{z}} \ r_m^{PH} \approx r_m^{HT} \end{cases} \tag{4}$$

In the aforementioned equations, the decision variable $\mathbf{z}$ represents the gene encoding enzymes, as determined using the NHDE algorithm for knockout. Fuzzy minimization (i.e., $\widetilde{\min}$) is used to evaluate the minimum cell growth rate and ATP production rate of the TR cells. By contrast, fuzzy maximization (i.e., $\widetilde{\max}$) is used to evaluate the maximum ATP production rate of the PH cells. Fuzzy dissimilarity (i.e., $\widetilde{\text{dissimilarity}}$) is used to determine the disparity between the fluxes (i.e., $v_j^{PH}$) and metabolite flow rates (i.e., $r_m^{PH}$) of the PH cells relative to those of the CA template. A substantial disparity indicates that the flux changes in the PH cells are considerably different from those in the CA template, indicating that the perturbation of HT cells cannot lead to tumorigenesis during treatment. Fuzzy similarity (i.e., $\widetilde{\text{similarity}}$) is used to evaluate the metabolic deviation between the PH cells and HT template. The flow rate of the $m^{th}$ metabolite is computed using the following equation:

$$r_m = \sum_{i \in \Omega^c} \left( \sum_{N_{ij}>0,j} N_{ij} v_{f,j} - \sum_{N_{ij}<0,j} N_{ij} v_{b,j} \right), m \in \Omega^m \tag{5}$$

where $\Omega^c$ is the set of metabolites located in various compartments of a cell and $N_{ij}$ is a stoichiometric coefficient of the $i^{th}$ metabolite in the $j^{th}$ reaction of a GSMM. The forward flux $v_{f,j}$ and backward flux $v_{b,j}$ of the $j^{th}$ reaction are calculated by applying FBA and UFD models to

the inner optimization problem as follows:

Treated CMS−specific model :

FBA problem :

$$\max_{\mathbf{v}_{f/b}} v_{biomass}$$

subject to

$$\mathbf{N}^{CA}(\mathbf{v}_f - \mathbf{v}_b) = \mathbf{0}$$

$$v_{f/b,i} = 0, i \in \Omega^{KO}$$

$$v_{f/b,j}^{LB} \leq v_{f/b,j} \leq v_{f/b,j}^{UB}, j \notin \Omega^{KO}$$

UFD problem :

$$\min_{\mathbf{v}_{f/b}} \sum_{k \in \Omega^{Int}} c_k^{CA}\left((v_{f,k})^2 + (v_{b,k})^2\right)$$

subject to

$$\mathbf{N}^{CA}(\mathbf{v}_f - \mathbf{v}_b) = \mathbf{0}$$

$$v_{f/b,i} = 0, i \in \Omega^{KO}$$

$$v_{f/b,j}^{LB} \leq v_{f/b,j} \leq v_{f/b,j}^{UB}, j \notin \Omega^{KO}$$

$$v_{biomass} \geq v_{biomass,CA}^*$$

Perturbed HT model :

FBA problem :

$$\max_{\mathbf{v}_{f/b}} v_{ATP}$$

subject to

$$\mathbf{N}^{HT}(\mathbf{v}_f - \mathbf{v}_b) = \mathbf{0}$$

$$v_{f/b,i} = 0, i \in \Omega^{KO}$$

$$v_{f/b,j}^{LB} \leq v_{f/b,j} \leq v_{f/b,j}^{UB}, j \notin \Omega^{KO}$$

UFD problem :

$$\min_{\mathbf{v}_{f/b}} \sum_{k \in \Omega^{Int}} c_k^{HT}\left((v_{f,k})^2 + (v_{b,k})^2\right)$$

subject to

$$\mathbf{N}^{HT}(\mathbf{v}_f - \mathbf{v}_b) = \mathbf{0}$$

$$v_{f/b,i} = 0, i \in \Omega^{KO}$$

$$v_{f/b,j}^{LB} \leq v_{f/b,j} \leq v_{f/b,j}^{UB}, j \notin \Omega^{KO}$$

$$v_{ATP} \geq v_{ATP,HT}^*$$

$$(6)$$

In the aforementioned equations, $\mathbf{N}^{CA}$ and $\mathbf{N}^{HT}$ are the stoichiometric matrices for each CMS-specific and HT model, respectively, and they are reconstructed from the models presented in the previous subsection; $v_{f/b,j}^{LB}$ and $v_{f/b,j}^{UB}$ are the positive lower bound (LB) and positive upper bound (UB) of the $j^{th}$ forward flux and $j^{th}$ backward flux, respectively. The RNA-seq expression data for the CA and HT cells and GPR associations in Recon3D were used to reconstruct GSMMs as well as to set the weighting factors $c_k^{CA}$ and $c_k^{HT}$ for the UFD problems; the four groups of confidence reactions are assigned as follows:

$$c_k^{CA/HT} = \begin{cases} \dfrac{1}{4}, k \in \text{high confidence} \\ \dfrac{1}{2}, k \in \text{medium confidence} \\ \dfrac{3}{4}, k \in \text{negativec confidence} \\ 1, k \in \text{other confidence or non−gene−expression} \end{cases} \quad (7)$$

For a high-confidence reaction, the weighting factor is set to the lowest value to enable a higher flux value to be obtained from the UFD problem. Forward and backward fluxes are set as zero if their corresponding gene encoding enzymes are knocked out. A reaction may be catalyzed by isozymes, which indicates that it is still active. GPR associations are used to set up the knockout reactions as follows:

$$v_{f,i} = v_{b,i} = 0; z_i \in \Omega^{KO} \backslash \Omega^{IZ}$$
$$\begin{cases} (1-\varepsilon)v_{f,i}^{basal} \leq v_{f,i} \leq (1+\varepsilon)v_{f,i}^{basal} \\ (1-\varepsilon)v_{b,i}^{basal} \leq v_{b,i} \leq (1+\varepsilon)v_{b,i}^{basal}; z_i \in \Omega^{KO} \cap \Omega^{IZ} \end{cases} \quad (8)$$

where $\Omega^{IZ}$ is a set of reactions regulated by isozymes represented in the GPR model.

## Maximizing decision-making problem

The ACTD problem expressed in Eqs (1) to (8) is a hierarchical multiobjective optimization problem (MOOP). Numerous methods have been employed for solving hierarchical MOOPs and to obtain a Pareto optimal solution [25, 49, 50]. These methods are generally classified into two categories: generating methods and preference-based methods [49, 50]. For generating methods, a scalarization approach is employed to convert a hierarchical MOOP into a single-objective optimization problem with multiple weighting factors to identify a Pareto optimal solution. By contrast, preference-based methods require a decision maker to indicate preferences in advance before a satisfactory solution can be identified. The present study used CA and HT templates (Fig 1D and 1E) as preferences to convert an ACTD problem into a maximizing decision-making (MDM) problem through the application of fuzzy set theory (Fig 1J and 1K). One-sided linear membership functions are applied to attribute fuzzy minimization (red line in Fig 1J) and fuzzy maximization (brown line in Fig 1J) as follows:

$$\eta_{\min} = \begin{cases} 1, & \text{if } FV < LB \\ \dfrac{UB - FV}{UB - LB}, & \text{if } LB \leq FV \leq UB \\ 0, & \text{if } FV > UB \end{cases}$$

$$\eta_{\max} = \begin{cases} 0, & \text{if } FV < LB \\ \dfrac{FV - LB}{UB - LB}, & \text{if } LB \leq FV \leq UB \\ 1, & \text{if } FV > UB \end{cases} \quad (9)$$

where $FV$ represents the flux values computed using the TR or PH model. The lower bound ($LB$) and upper bound ($UB$) are obtained using the corresponding CA and HT templates, (i.e., $LB = ST/4$ and $UB = 4ST$; $ST$ is the standard value for a CA or HT template) used in the present study. Two-sided linear membership functions are used to attribute fuzzy dissimilarity (blue line in Fig 1J) and fuzzy similarity (green line in Fig 1J). Fuzzy dissimilarity is a complement of fuzzy similarity; therefore, the fuzzy similarity grade is derived using the

equation as follows:

Left−hand side membership function :

$$\eta_L^{PHST} = \begin{cases} 0, & \text{if } FV < LB \\ \dfrac{FV - LB}{ST - LB}, & \text{if } LB \leq FV \leq ST \\ 1, & \text{if } FV = ST \end{cases}$$

Right−hand side membership function :

$$\eta_R^{PHST} = \begin{cases} 1, & \text{if } FV = ST \\ \dfrac{UB - FV}{UB - ST}, & \text{if } ST \leq FV \leq UB \\ 0, & \text{if } FV > UB \end{cases}$$

(10)

The fuzzy similarity grade can be calculated using the equation $\eta_{SI} = \max\{\min\{\eta_L^{PHST}, \eta_R^{PHST}, 1\}, 0\}$, and its complement, fuzzy dissimilarity, can be obtained using the equation $\eta_{DS} = 1 - \eta_{SI}$.

The ACTD problem in Eqs (1) to (8) can be transformed into an MDM problem by applying the membership functions as follows:

$$\begin{cases} \max_{\mathbf{z}} \eta_D = \max_{\mathbf{z}}(\eta_{TR} + \min\{\eta_{TR}, \eta_{CV}, \eta_{MD}\})/2 \\ \text{subject to inner optimization problems} \\ 1.\text{FBA and UFD problems for treated CMS cells} \\ 2.\text{FBA and UFD problems for perturbed HT cells} \end{cases}$$

(11)

where the decision objective $\eta_D$ is a hierarchical criterion, and the cell mortality grade $\eta_{TR}$ is used to achieve the first objective of Eq (1) with respect to the outer optimization problem and is considered the first priority in the fuzzy decision-making problem. The cell viability grade $\eta_{CV}$ is applied to achieve the second objective of Eq (2), and the metabolic deviation grade $\eta_{MD}$ is used to achieve the third and fourth goals. The second priority of the decision objective is used to evaluate the lowest grade in the set $\{\eta_{TR}, \eta_{CV}, \eta_{MD}\}$ when the cell viability or metabolic deviation grade is less than the cell mortality grade. We introduce a mean-min operation to compute the cell mortality grade ($\eta_{TR}$), cell viability grade ($\eta_{CV}$), and metabolic deviation grade ($\eta_{MD}$) as follows:

$$\eta_{TR} = ((\eta_{ATP}^{TR} + \eta_{biomass}^{TR})/2 + \min\{\eta_{ATP}^{TR}, \eta_{biomass}^{TR}\})/2$$

(12)

$$\eta_{CV} = ((\eta_{ATP}^{PH} + \eta_{biomass}^{PH})/2 + \min\{\eta_{ATP}^{PH}, \eta_{biomass}^{PH}\})/2$$

(13)

$$\eta_{MD} = ((\eta_{DS} + \eta_{SI})/2 + \min\{\eta_{DS}, \eta_{SI}\})/2$$

(14)

where the membership grades ($\eta_{ATP}^{TR}, ..., \eta_{SI}$) are obtained from the membership functions defined in Eqs (9) and (10).

The MDM problem of Eq (11) is a mixed-integer optimization problem with linear and quadratic programming problems in its inner loop. It is a high-dimensional, nondeterministic, polynomial-time hard problem that cannot be solved using currently available commercial software. We employed the NHDE algorithm to solve this MDM problem. The NHDE

algorithm is a parallel direct search procedure and an extension of the hybrid differential algorithm [51]. The computational procedures are detailed in the supporting information (S1 Text). The source programs of the ACTD platform for identifying the anticancer genes, metabolites, and reactions of CMSs are available at http://doi.org/10.5281/zenodo.7136561.

## Results

### CMS-specific metabolic models

We used RNA-seq expression data for each CMS and its HT counterpart to reconstruct corresponding CMS-specific and healthy GSMMs (Fig 1A–1C). Fig 3 presents the numbers of metabolites, reactions, genes, and feasible encoded enzymes for each reconstructed model. As indicated by the blue regions in Fig 3, five CMSs and HT models shared numerous similarities in terms of their metabolites, reactions, genes, and enzymes. The orange regions indicate additional shared items in the five CMSs, and the grey regions illustrates the items that are specific to each CMS and HT model. Fig 4 illustrates the top 10 metabolites and reaction classifications for each CMS and HT model. Most metabolites in the fatty acyl groups and high percentages of organooxygen compounds, carboxylic acids, and steroids are shared by the CMSs and HT model (Fig 4A). The metabolites that are specific to CMS2 to CMS5 comprise numerous steroid derivatives. Furthermore, extracellular transport reactions accounted for the highest percentage of reactions in the CMSs and HT model. More than 800 fatty acid oxidation reactions were shared by the CMSs and HT model (Fig 4B).

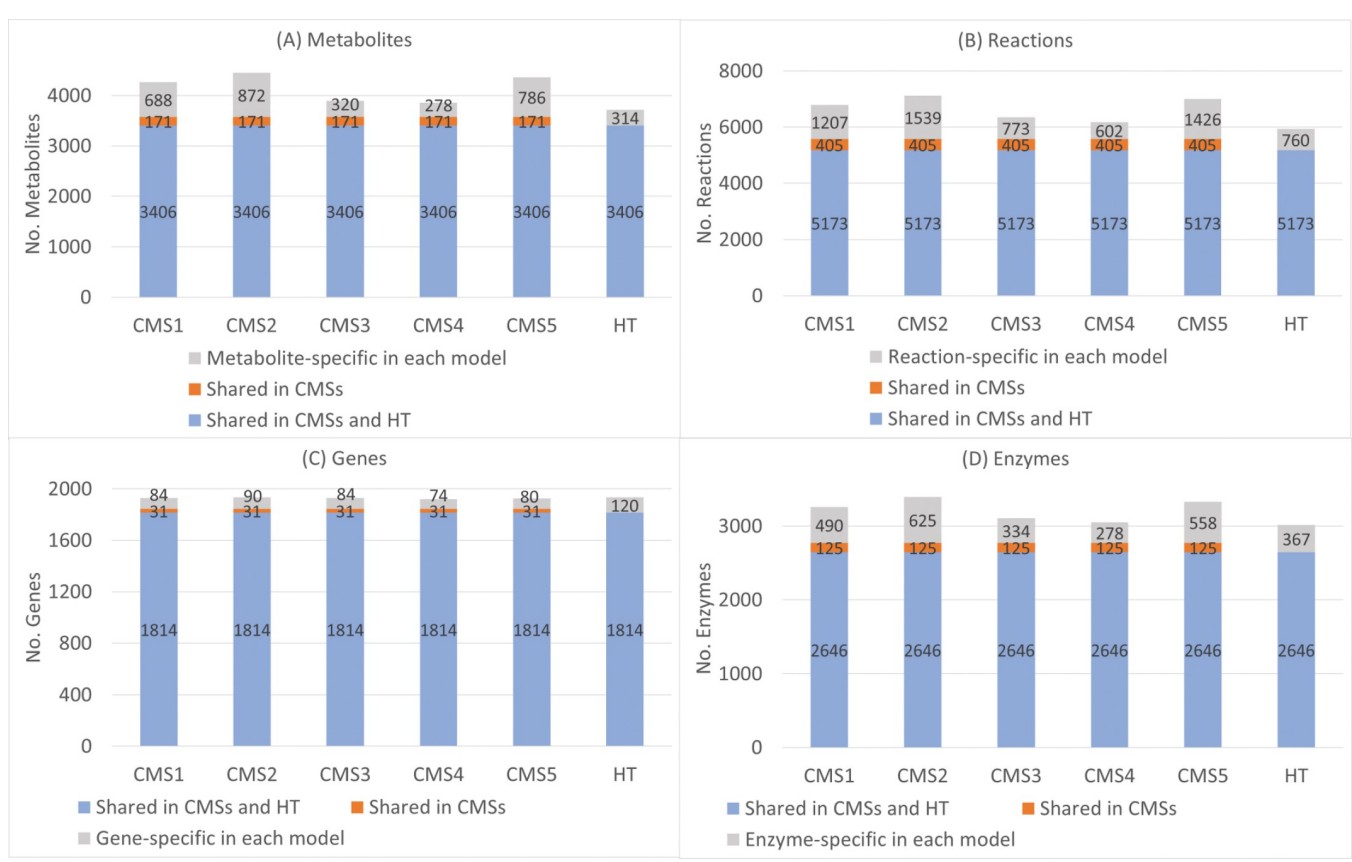

**Fig 3. Number of metabolites, reactions, genes, and feasible enzymes for each consensus molecular subtype and its healthy counterpart.**

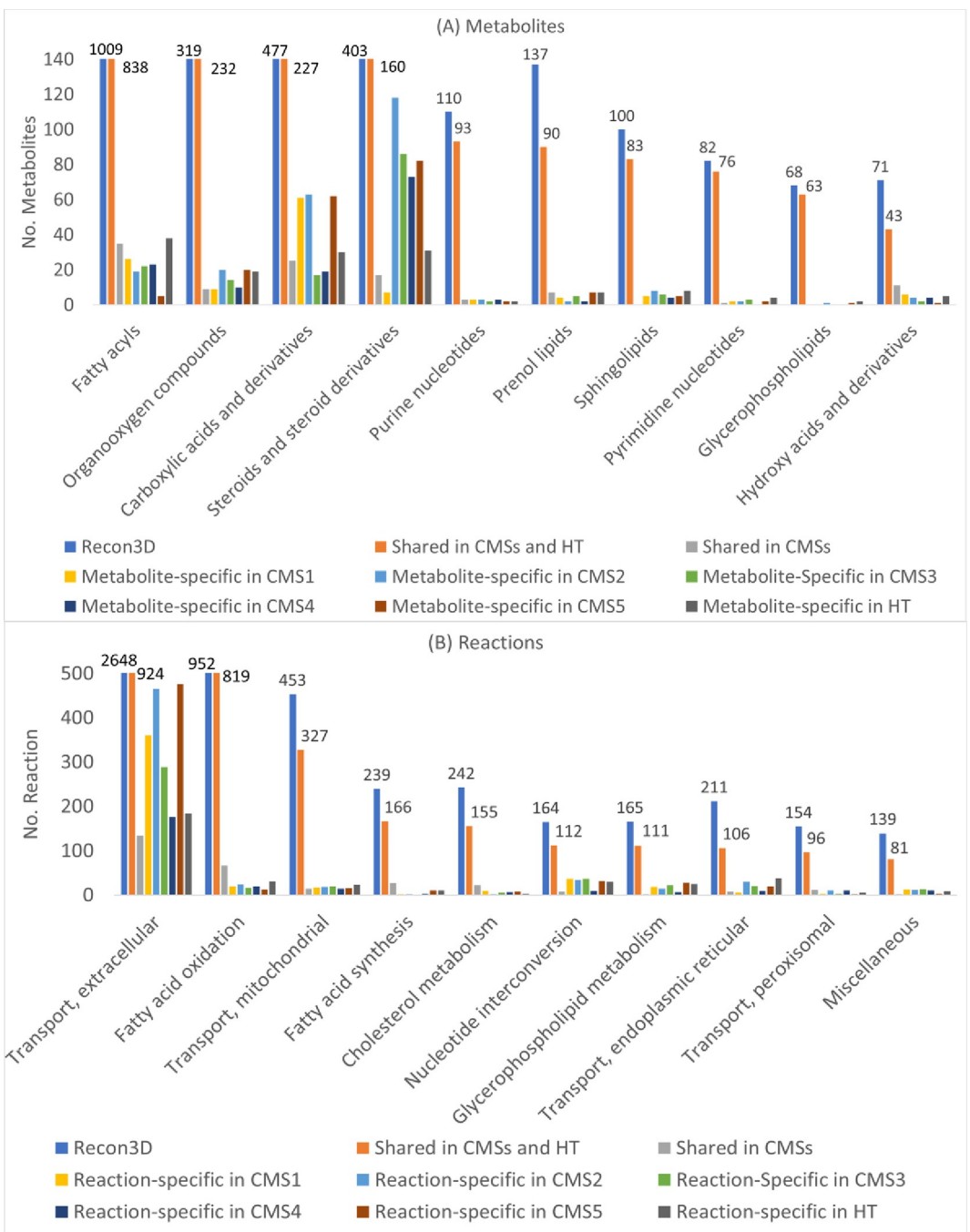

**Fig 4. Classification of species and reactions in Recon3D, consensus molecular subtypes and its healthy counterpart.**
The numbers above the blue and orange bars represent the total numbers of metabolites and reactions for Recon3D and that of those shared between the CMSs and HT models.

## Identifying single essential genes

We used Dulbecco's Modified Eagle Medium (DMEM) (S1 Table) and set 51 uptake reactions in DMEM as reversible exchangeable reactions in our computations. Secretion reactions for each model were set as irreversible reactions. The NHDE algorithm [25–29, 49] was used to perform a series of computations to solve the MDM problem and identify a set of essential

**Table 3. Cell mortality grades and metabolic deviation grades of essential genes for each consensus molecular subtype, obtained using DMEM medium.** N/D is obtained from the DepMap portal (https://depmap.org/portal/) and is defined as the ratio of the cell death number (N) and the total number of colon cancer cell lines (D) used in the experimental test. "No.Drugs" denotes the number of drugs retrieved from DrugBank (https://go.drugbank.com/) that modulate each gene. "SE.Score" denotes the side effect score, calculated using average adverse events. "—" indicates that data were not available on the databases. "*" indicates that ADSS2 and CTPS1 are the representative of duplicate enzymes (ADSS1 and ADSS2) and (CTPS1 and CTPS2), respectively.

| Gene | Cell Mortality Grade | | | | | Metabolic Deviation Grade | | | | | N/D | No. Drugs | SE. Score |
|------|------|------|------|------|------|------|------|------|------|------|-----|-----------|-----------|
| | CMS1 | CMS2 | CMS3 | CMS4 | CMS5 | CMS1 | CMS2 | CMS3 | CMS4 | CMS5 | | | |
| DHODH | 0.660 | 0.705 | 0.670 | 0.606 | 0.691 | 0.658 | 0.682 | 0.498 | 0.404 | 0.508 | 17/54 | 26 | 0.351 |
| CAD | 0.660 | 0.705 | 0.670 | 0.606 | 0.691 | 0.654 | 0.696 | 0.508 | 0.385 | 0.500 | 27/54 | 3 | -- |
| RPIA | 0.660 | 0.705 | 0.670 | 0.606 | 0.691 | 0.638 | 0.611 | 0.549 | 0.512 | 0.516 | 6/54 | 1 | -- |
| SQLE | 0.660 | 0.705 | 0.670 | 0.606 | 0.691 | 0.575 | 0.693 | 0.548 | 0.489 | 0.488 | 2/54 | 4 | 0.415 |
| ADSS2* | 0.660 | 0.705 | 0.670 | 0.606 | 0.691 | 0.569 | 0.680 | 0.464 | 0.499 | 0.504 | 31/54 | 2 | -- |
| EBP | 0.660 | 0.705 | 0.670 | 0.606 | 0.691 | 0.568 | 0.668 | 0.450 | 0.566 | 0.504 | 0/54 | 1 | 0.412 |
| TM7SF2 | 0.660 | 0.705 | 0.670 | 0.606 | 0.691 | 0.558 | 0.702 | 0.496 | 0.496 | 0.507 | 1/54 | -- | -- |
| MVD | 0.660 | 0.705 | 0.670 | 0.606 | 0.691 | 0.557 | 0.628 | 0.509 | 0.480 | 0.495 | 49/54 | -- | -- |
| MVK | 0.660 | 0.705 | 0.670 | 0.606 | 0.691 | 0.557 | 0.628 | 0.509 | 0.482 | 0.495 | 52/54 | 1 | -- |
| PMVK | 0.660 | 0.705 | 0.670 | 0.606 | 0.691 | 0.557 | 0.681 | 0.509 | 0.480 | 0.495 | 19/54 | -- | -- |
| LSS | 0.660 | 0.705 | 0.670 | 0.606 | 0.691 | 0.556 | 0.693 | 0.547 | 0.488 | 0.489 | 0/54 | 2 | -- |
| UMPS | 0.660 | 0.705 | 0.670 | 0.606 | 0.691 | 0.554 | 0.547 | 0.546 | 0.470 | 0.532 | 29/54 | 2 | -- |
| FDFT1 | 0.660 | 0.705 | 0.670 | 0.606 | 0.691 | 0.553 | 0.617 | 0.494 | 0.496 | 0.503 | 7/54 | 1 | -- |
| HMGCR | 0.660 | 0.705 | 0.670 | 0.606 | 0.691 | 0.543 | 0.625 | 0.469 | 0.485 | 0.544 | 52/54 | 20 | 0.413 |
| CRLS1 | 0.660 | 0.705 | 0.670 | 0.606 | 0.691 | 0.539 | 0.667 | 0.469 | 0.572 | 0.494 | 35/54 | -- | -- |
| SLC7A6 | 0.657 | 0.704 | 0.668 | 0.605 | 0.690 | 0.536 | 0.598 | 0.443 | 0.491 | 0.485 | 0/54 | 1 | -- |
| PGS1 | 0.660 | 0.705 | 0.670 | 0.606 | 0.691 | 0.488 | 0.596 | 0.482 | 0.498 | 0.501 | 52/54 | -- | -- |
| SC5D | 0.660 | 0.705 | 0.670 | 0.606 | 0.691 | 0.485 | 0.696 | 0.486 | 0.503 | 0.501 | 0/54 | -- | -- |
| ADSL | 0.660 | 0.705 | 0.670 | 0.606 | 0.691 | 0.470 | 0.701 | 0.469 | 0.502 | 0.517 | 51/54 | -- | -- |
| NSDHL | 0.660 | 0.705 | 0.670 | 0.606 | 0.691 | 0.433 | 0.588 | 0.526 | 0.489 | 0.499 | 0/54 | 1 | -- |
| SLC2A13 | 0.660 | -- | -- | 0.606 | -- | 0.448 | -- | -- | 0.528 | -- | 0/54 | -- | -- |
| CYP51A1 | 0.660 | -- | -- | -- | 0.691 | 0.435 | -- | -- | -- | 0.495 | 0/54 | 5 | 0.359 |
| CTPS1* | -- | 0.705 | -- | -- | 0.691 | -- | 0.653 | -- | -- | 0.495 | 44/54 | 1 | -- |
| SLC5A3 | -- | -- | 0.670 | -- | -- | -- | -- | 0.541 | -- | -- | 2/54 | 1 | -- |
| PTPMT1 | -- | -- | -- | 0.606 | -- | -- | -- | -- | 0.502 | -- | 28/54 | -- | -- |
| PTDSS1 | -- | -- | -- | 0.606 | -- | -- | -- | -- | 0.488 | -- | 8/54 | 1 | -- |

genes for each CMS (Table 3). The computational procedures of the NHDE algorithm are described in the supporting information (S1 Text). The performance and solution quality of the NHDE algorithm are dependent on three key setting factors: the tolerance ratio used in migration, population size, and maximum number of iterations. In the present study, a tolerance ratio of 0.05, a population size of 50, and 50 maximum iterations were used. Moreover, 10 random seeds were used for each run to obtain optimal solutions.

Through the computation, we identified 26 essential genes for the CMSs (Table 3), of which 20 were shared by the CMSs and 6 were CMS-specific. We also used a brute-force enumeration algorithm to individually identify essential genes to validate our computations, and the results were identical to those obtained using the NHDE algorithm. With 3.7 GHz processor and 64 GB RAM on an i9 local computer, the enumeration method demanded ~8 CPU hours, while the NHDE algorithm ~6 CPU hours. STRING (https://string-db.org/) and GeneCards https://www.genecards.org/) were used to classify the protein–protein interaction (PPI) networks encoded by the 26 genes into five classes (Fig 5A). The first class comprised 12 genes involved in cholesterol biosynthesis. The second class comprised six genes that participate in nucleotide metabolism (specifically purine and pyrimidine metabolism) and one gene involved in the

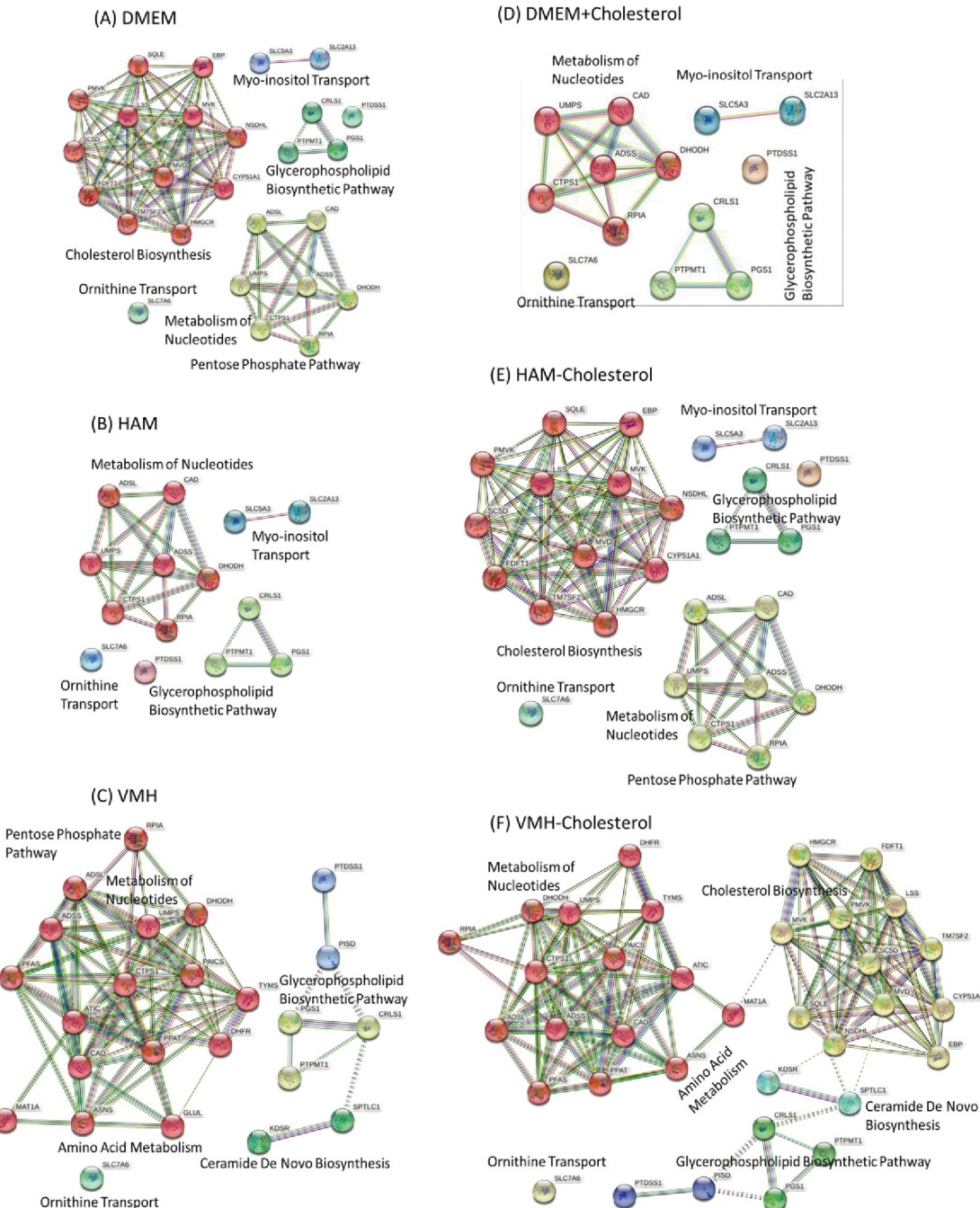

**Fig 5. Protein–protein interactions of identified essential genes for the union set of five consensus molecular subtypes in various media.** (A) DMEM: Dulbecco's Modified Eagle Medium. (B) HAM: Ham's medium. (C) VMH: Uptake reactions obtained from the VMH database (https://www.vmh.life/#home). (D) DMEM+Cholesterol: DMEM medium with cholesterol uptake. (E) HAM-Cholesterol: Ham's medium without cholesterol uptake. (F) VMH-Cholesterol: VMH medium without cholesterol uptake.

pentose phosphate pathway. The third class comprised four genes involved in the glycerophospholipid biosynthetic pathway and other myo-inositol and ornithine transporters.

The computational results may be affected by different weighting factors assigned by Eq (6). In this study, we also considered two additional weighting factors, i.e. the weighting factors

based on gene expression levels from zero to one and equal weighting factors, to compute the corresponding flux distributions, and compared them with that using Eq (7). The ACTD platform used the DMEM medium and different weighting factors to identify a set of single essential genes as shown in S2 Table, respectively. The results reveal that the identified essential genes are irrespective to the assigned weighting factors (S2 Table). However, the cell mortality grades and metabolic deviation grades are different a little. An essential gene is influenced by the topology of a metabolic network, that is a stoichiometric matrix reconstructed from gene expression data. However, weighting factors could make to obtain different flux values, not influence the network structure.

For each essential gene, the proliferation of cancer cells under all CMS models could be terminated (cell growth rate $\leq 10^{-8}$), and the ATP production rate of these cells decreased by approximately 60% relative to their maximum levels. As a result, the cell mortality grade of the cancer cells was more than 0.6 (Table 3), indicating that the first fuzzy objective of Eq (1) was achieved with a 60% satisfaction level. On the basis of this computation, we assumed that the genes in the HT cells were also blocked and investigated the cell viability and metabolic deviation of the PH cells. We discovered a cell maintenance rate of $10^{-8}$ in the PH cells and that these cells had the highest ATP production rate; thus, the second fuzzy objective was achieved with a 100% satisfaction level. The metabolic deviation grade $\eta_{MD}$ was used to perform mean–min calculations to achieve the third and fourth objectives of Eq (1), that is, to evaluate the flux perturbation and metabolite flow rates in the PH cells relative to those of the CA and HT templates, respectively. For the CMS1 model, the highest metabolic deviation grade ($\eta_{MD}$) of 0.658 was achieved with the knockout of dihydroorotate dehydrogenase DHODH, and the lowest $\eta_{MD}$ value of 0.433 was achieved with the knockout of sterol-4-alpha-carboxylate 3-dehydrogenase NSDHL. A higher grade was considered to indicate fewer predicted metabolic perturbations. For the models for CMS2 to CMS5, the height of the metabolic deviation grade varied with the gene that was knocked out (e.g., TM7SF2, RPIA, CRLS1, and HMGCR; Table 3).

A survey of a cancer dependency map (DepMap, https://depmap.org/portal/) revealed that most of the identified genes were compatible with the colon cancer cell lines obtained from DepMap and that the knockout of these genes (except for SLC7A6, SC5D, LSS, EBP, NSDHL, and SLC2A13) resulted in a high amount of cell death (Table 3). Some of the identified essential genes can be modulated using drugs that have been approved and are listed on DrugBank [52]. For example, 26 and 20 of the drugs on DrugBank that were surveyed can modulate the expression of DHODH and HMGCR, respectively [52]. To investigate the grades of adverse events (AEs), the aforementioned approved drugs were used in a SIDER survey (http://sideeffects.embl.de/) conducted using the ADDReSS (http://www.bio-add.org/ADReCS/) database. The National Cancer Institute Common Terminology Criteria for Adverse Events provides precise clinical descriptions of AE severity ranging from mild to associated with death and grades AE on a scale of 1, 2, 3, 4, and 5. We used the scale to calculate the average AE (Ave.AE) and converted it into a side effect score (i.e., SE.Score = 1 − Ave.AE) that correlates with the corresponding metabolic deviation grade (Table 3).

## Uptake reactions

We used Ham's medium as a nutrient, and the 63 uptake reactions (S1 Table) that occurred in this medium were used to identify the essential genes for each CMS. The computational results (S3 Table) revealed that the cell viability and metabolic deviation grades obtained using Ham's medium were almost identical to those obtained using DMEM. The union set of CMSs comprised 14 identified essential genes, of which nine were shared by CMSs and five were CMS-specific. The union set was used to categorize PPIs into four classes (Fig 5B), and the results

revealed that the identified essential genes are not involved in the cholesterol biosynthetic pathway. The 14 essential genes identified using Ham's medium form a subset of those identified using DMEM. We discovered that six genes (UMPS, CAD, DHODH, ADSS2, ADSL, and CTPS1) participate in purine and pyrimidine metabolism, RPIA participates in the pentose phosphate pathway, four genes (PGS1, CRLS1, PTPMT1, and PTDSS1) participate in the glycerophospholipid biosynthetic pathway, SLC7A6 participates in ornithine transport, and two genes (SLC5A3 and SLC2A13) participate in the nuclear receptor meta-pathway.

The VMH database (https://www.vmh.life/#home) published data on 91 uptake reactions that are involved in human nutrient uptake. We used the data of these uptake reactions to investigate the effects of such reactions on essential genes. Some of the uptake reactions were not present in the reconstructed GSMMs of the CMSs; thus, they were excluded from the computations. In total, 83 uptake reactions (S1 Table) were included in the computations. The NHDE algorithm revealed that 15 essential genes were shared by the CMSs and 9 genes were CMS-specific (S3 Table). The union set of the 24 essential genes of the CMSs was used to distinguish the PPIs into four classes (Fig 5C), and the results revealed that the identified essential genes are not involved in the cholesterol biosynthetic pathway. The aforementioned findings indicate that these genes are nonessential when Ham's medium or VMH medium are used because the cancer cells of all CMSs continue to survive if the genes involved in cholesterol biosynthesis are knocked out.

A comparison of the uptake reactions for the DMEM, Ham's medium, and VMH revealed that no cholesterol uptake reaction occurred when DMEM was used. We used three additional media to investigate the relationship of tumor cell growth with nutrient components and essential genes. Therefore, DMEM was used to create another medium in which a cholesterol uptake reaction was induced (DMEM+Cholesterol). Ham's medium and VMH medium in which a cholesterol uptake reaction was not induced (denoted as the HAM-Cholesterol and VMH-Cholesterol, respectively) were used as the second and third media, respectively. On the ACTD platform, each medium was used to identify the essential genes for each CMS. Fig 5D–5F illustrates the PPI networks of the union set of essential genes that were identified using the three additional media. The results are presented in S3 Table. When the DMEM+Cholesterol was used, the essential genes involved in the cholesterol biosynthetic pathway could not be determined (Fig 5D). However, the essential genes in the cholesterol biosynthetic pathway could be identified using the HAM-Cholesterol and VMH-Cholesterol. This finding reveals that the essential genes in the cholesterol biosynthetic pathway can be determined if a cholesterol uptake reaction is not induced in media (Fig 5A, 5E, and 5F). The genes in the cholesterol biosynthetic pathway became nonessential if a cholesterol uptake reaction was induced in a medium (Fig 5B–5D). Our simulation results are consistent with those reported by other studies [38–40]; that is, medium components can influence cell growth associated with tumor metabolism. We additionally set all exchange reactions (≥782 reactions) for each CMS as reversible uptake reactions to enable identification of essential genes; the results indicated that only CRLS1 was determinable for all CMSs, indicating that CRLS1 is a medium-independent essential gene.

The metabolite flow distributions for CMS1 that were evaluated using the DMEM and the DMEM+Cholesterol are illustrated in Fig 6. These distribution findings explained the relationship between the cholesterol uptake reactions and essential genes; the HT cells metabolized glucose to pyruvate with a flux of 13.649 mmol/gDW h (first column in the data box of Fig 6) when DMEM was used and with a flux of 13.650 mmol/gDW h (second column in the data box of Fig 6) when the DMEM+Cholesterol was used. Pyruvate then entered the TCA cycle to generate ATP, which is required for cell survival. The Warburg hypothesis posits that cancer cells rewire their metabolism to promote growth, survival, proliferation, and long-term

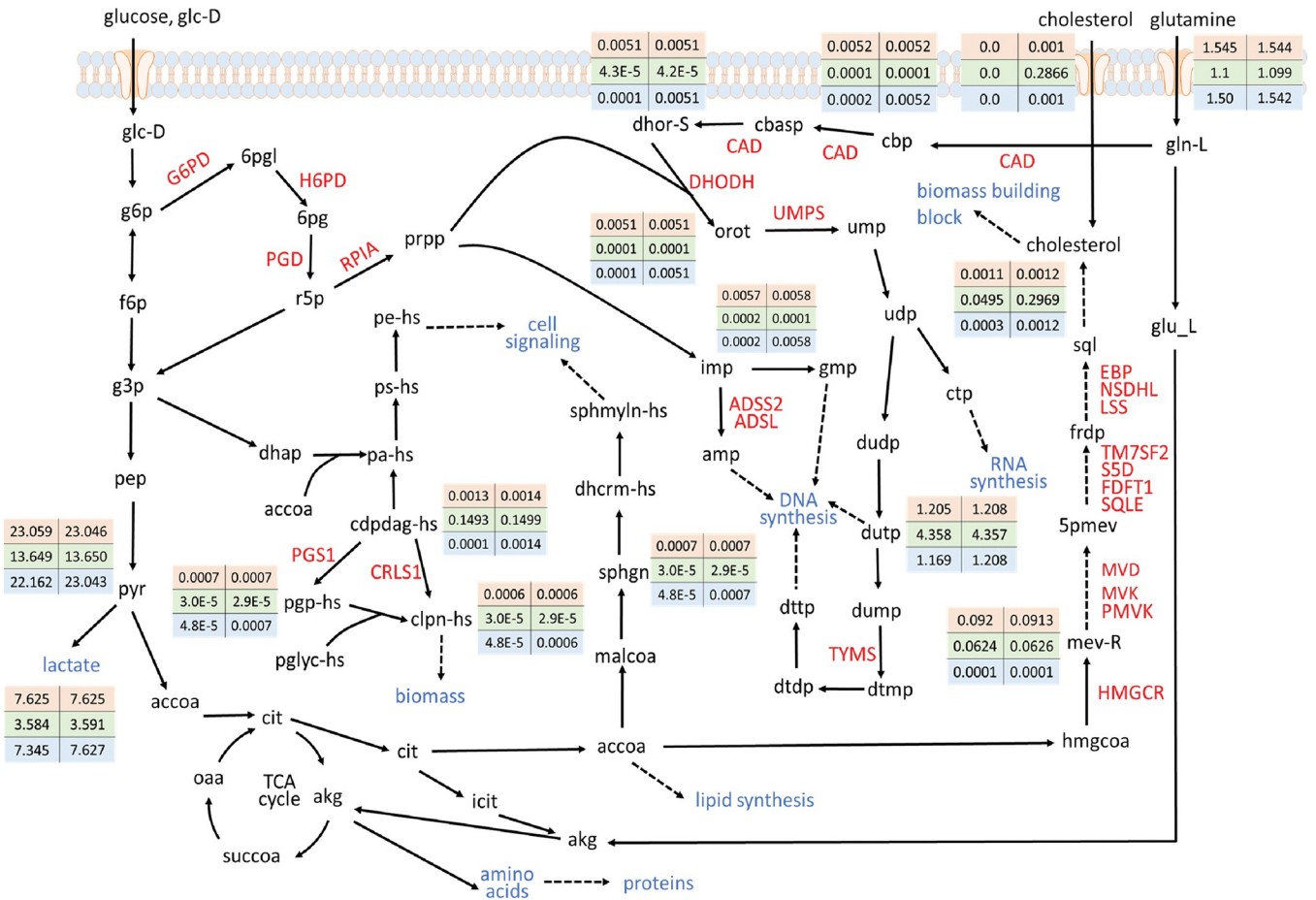

**Fig 6. Metabolite flow rates for CMS1 obtained using DMEM (first column in matrix of data) and DMEM+Cholesterol (second column).** The first column in each data box presents the computational results obtained using DMEM; the second column presents the results obtained using the DMEM +Cholesterol. The first and second rows of each data box present the CA and HT templates, respectively; the third row presents the genes that are knocked out in the cholesterol biosynthetic pathway (e.g., HMGCR and PMVK).

maintenance. Through the CMS1 model, we discovered that the CA template metabolizes large amounts of glucose to pyruvate (23.059 for DMEM and 23.046 for DMEM+Cholesterol), which is then mostly converted to lactic acid (7.625 for both media). Thereafter, glutamine is replenished in the cycle to meet the proliferation requirement. These results are consistent with the Warburg hypothesis [53–55]. We blocked HMGCR and other genes, such as PMVK and MVK in the cholesterol biosynthetic pathway (Fig 6) to evaluate specific metabolite flow rates (listed in the third row of data boxes in Fig 6). The results revealed that the metabolite flow rate of mev-R was 0.0001 for both DMEM and DMEM+Cholesterol and that a series reaction produced intracellular cholesterol at a level of 0.0003 for DMEM and of 0.0012 for DMEM+Cholesterol. Cholesterol production occurred not only because of intracellular biosynthesis but also because of an uptake reaction induced by an extracellular medium. No extracellular cholesterol uptake was observed when DMEM was used (Fig 6) because this uptake reaction was not induced in the medium. This rendered the genes in the cholesterol biosynthetic pathway essential. By contrast, when the DMEM+Cholesterol was used, the deletion of HMGCR prevented intracellular biosynthesis; however, cholesterol was still supplied at a level of 0.001 through the extracellular uptake reaction that occurred in the medium. Consequently, HMGCR became a nonessential gene when the DMEM+Cholesterol was used.

## Combination of essential genes

Conducting an enumerative search for identifying combinations of two target essential genes is time consuming because it requires computations to be performed for more than 4,800,000 combinations for each CMS. We are difficult to use an enumerative search for identifying combinations of two targets. The ACTD platform can be used to reduce the computational burden associated with performing evolutionary procedures. Accordingly, we employed the ACTD platform to identify gene combinations for various uptake reactions (S1 Table) by using two candidate groups obtained using the NHDE algorithm. The first candidate group comprised the union set of essential genes shared by CMSs (Table 3), and the second group comprised the other candidate genes from feasible encoding enzymes. This strategy substantially reduced the computational time and reduced the search space to approximately 86,000 possible combinations for the two candidate groups. Our computational results revealed that the metabolic deviation grades for most two-target combinations (S4 Table) were superior to those for their corresponding one-target essential genes (S3 Table) and that each combination involved at least one essential gene. Moreover, the results for the one-target essential genes presented in Fig 5 indicate that a medium in which cholesterol uptake occurs cannot be used to identify a one-target essential gene in the cholesterol biosynthetic pathway; however, a one-target essential gene can be combined with another essential gene to improve the metabolic deviation grade of the combination.

## Essential metabolites and reactions

The presence of essential metabolites and reactions indicates that cancer cells will terminate to grow and normal cells will survive if all metabolite synthesis reactions are blocked. We applied metabolite- and reaction-centric approaches on the ACTD platform to identify essential metabolites and reactions, respectively, for various media (S1 Table). We compared the computational results between various media with or without cholesterol uptake reactions and discovered that most of the essential metabolites were shared by CMSs irrespective of the presence or absence of a cholesterol uptake reaction (S5 Table). The computational results also revealed that the essential metabolites farnesyl diphosphate (abbreviated as frdp in Fig 6) and formate can be identified in a medium without a cholesterol uptake reaction. The only exception to this occurred in the CMS5 model with VMH-Cholesterol. Farnesyl diphosphate is a chemical compound in prenol lipids that participates in the mevalonate metabolic pathway, and it is catalyzed by FDFT1 to form squalene (sql) and to progressively synthesize cholesterol (Fig 6). Formate is a crucial molecule in one-carbon metabolism that it serves as an intermediate in is a series of biochemical reactions, including the biosynthesis of cholesterol, nucleotides and amino acids [56]. Specifically, formate is used as a one-carbon donor in the conversion of acetyl-CoA to mevalonate, which is a key intermediate in the cholesterol biosynthetic pathway. In the present study, the inhibition of formate production rate reduced intracellular cholesterol synthesis and thereby eliminated the growth of cancer cells.

The essential reactions for all CMSs were identified using various media (S1 Table). No essential reaction was identified in the cholesterol biosynthetic pathway irrespective of whether a cholesterol uptake reaction occurred in a medium (S6 Table). This finding differs from that obtained using a gene-centric approach (Table 3). This different finding may have occurred because each essential gene that is involved in the cholesterol biosynthetic pathway through GPR association can regulate at least two reactions. We used two-reaction combinations catalyzed by HMGCR to compute the cell mortality and metabolic deviation grades, and the results were consistent with those obtained when HMGCR was knocked out. Table 4 reveals that the reactions R_DHORD9, R_ADSL1, and R_ADSS were regulated by their corresponding genes,

**Table 4. Cell mortality grades and metabolic deviation grades of essential reactions for each consensus molecular subtype, obtained using DMEM.** Setting the flux value of an essential reaction to zero leads to termination of cancer cell growth and proliferation.

| | Cell Mortality Grade | | | | | Metabolic Deviation Grade | | | | | |
|---|---|---|---|---|---|---|---|---|---|---|---|
| ID | CMS1 | CMS2 | CMS3 | CMS4 | CMS5 | CMS1 | CMS2 | CMS3 | CMS4 | CMS5 | Name |
| R_ASPCTr | 0.660 | 0.705 | 0.670 | 0.606 | 0.691 | 0.663 | 0.632 | 0.520 | 0.504 | 0.532 | Aspartate Carbamoyltransferase |
| R_DHORD9 | 0.660 | 0.705 | 0.670 | 0.606 | 0.691 | 0.658 | 0.682 | 0.498 | 0.404 | 0.508 | Dihydoorotic Acid Dehydrogenase |
| R_OMPDC | 0.660 | 0.705 | 0.670 | 0.606 | 0.691 | 0.655 | 0.555 | 0.530 | 0.499 | 0.526 | Orotidine-5'-Phosphate Decarboxylase |
| R_ORPT | 0.660 | 0.705 | 0.670 | 0.606 | 0.691 | 0.618 | 0.687 | 0.524 | 0.496 | 0.532 | Orotate Phosphoribosyltransferase |
| R_CBPS | 0.660 | 0.705 | 0.670 | 0.606 | 0.691 | 0.594 | 0.527 | 0.422 | 0.513 | 0.500 | Carbamoyl-Phosphate Synthase |
| R_ADSL1 | 0.660 | 0.705 | 0.670 | 0.606 | 0.691 | 0.569 | 0.680 | 0.464 | 0.499 | 0.504 | Adenylosuccinate Lyase |
| R_ADSS | 0.660 | 0.705 | 0.670 | 0.606 | 0.691 | 0.569 | 0.680 | 0.464 | 0.499 | 0.504 | Adenylosuccinate Synthase |
| R_DHORTS | 0.660 | 0.705 | 0.670 | 0.606 | 0.691 | 0.552 | 0.551 | 0.497 | 0.513 | 0.520 | Dihydroorotase |
| R_DATPtn | 0.660 | 0.705 | 0.670 | 0.606 | 0.691 | 0.504 | 0.681 | 0.498 | 0.528 | 0.547 | DATP Diffusion in Nucleus |
| R_DGTPtn | 0.660 | 0.705 | 0.670 | 0.606 | 0.691 | 0.459 | 0.702 | 0.532 | 0.574 | 0.493 | DGTP Diffusion in Nucleus |
| R_DSAT | 0.660 | 0.705 | 0.670 | 0.606 | 0.691 | 0.449 | 0.677 | 0.452 | 0.503 | 0.546 | Dihydrosphingosine N-Acyltransferase |
| R_INSTt2r | 0.660 | -- | -- | 0.606 | -- | 0.448 | -- | -- | 0.528 | -- | Transport of Inositol via Proton Symport |
| R_TRDR | 0.660 | -- | 0.670 | -- | -- | 0.510 | -- | 0.545 | -- | -- | Thioredoxin Reductase (NADPH) |
| R_INSTt4 | -- | -- | 0.670 | -- | -- | -- | -- | 0.534 | -- | -- | Transport of Inositol via Sodium Symport |
| R_PEt | -- | -- | 0.510 | -- | 0.560 | -- | -- | 0.512 | -- | 0.477 | Phosphatidylethanolamine Transport |
| R_PGPP_hs | -- | -- | -- | 0.606 | -- | -- | -- | -- | 0.502 | -- | Phosphatidylglycerol Phosphate Phosphatase |
| R_PSSA1_hs | -- | -- | -- | 0.606 | -- | -- | -- | -- | 0.488 | -- | Phosphatidylserine Synthase |

namely DHODH, ADSL, and ADSS1, respectively. Consequently, the blockage of each reaction matched with the corresponding gene knockout (Table 4). The essential gene CAD (Table 3) catalyzed three sequential reactions, namely R_CBPS, R_ASPCTr, and R_DHORTS. Therefore, the blockage of each reaction inhibited the growth and proliferation of the cancer cells of each CMS and yielded a satisfactory metabolic deviation grade (Table 4). The essential gene UMPS regulated two sequential reactions, namely R_OMPDC and R_ORPT, in the pyrimidine metabolic pathway, and similar results were obtained.

## Discussion

CRC is a major disease burden worldwide, and improved prognoses and treatment strategies are urgently required. The development of molecular subtype–based therapies has provided a new potential framework for implementing preferred and precise medical treatments. CRC samples obtained from patients can be categorized into five subtypes through CMS classification, which is based on RNA expression in CRC. Several studies have used CMS classification to predict a patient's prognosis and determine treatment strategies for CRC. However, few studies have assessed the use of CMS classification to reconstruct CMS-specific GSMMs and to analyze the metabolic characteristics of these GSMMs. The present study used RNA-seq expression data of CRC retrieved from TCGA to reconstruct five CMS-specific GSMMs. We discovered that the five CMSs and HT model had numerous similarities with respect to metabolites, reactions, genes, and enzymes. The CMSs and HT shared the most metabolites in the fatty acyl groups and more than 800 fatty acid oxidation reactions.

The proposed fuzzy hierarchical optimization framework can be used to identify essential genes, metabolites, and reactions for treating each CMS of CRC. The optimization framework can be applied to identify essential targets that lead to termination of cancer cell growth and to evaluate metabolic flux perturbations in normal cells caused by cancer treatment. In addition, metabolic deviation grades and two-sided fuzzy membership functions were used to evaluate

the flux perturbations and metabolite flow rates in perturbed HT cells relative to those in HT and CA templates, respectively. A smaller metabolic deviation was considered to indicate fewer adverse effects. We used various media to identify essential targets for each CMS and discovered that most targets were shared by the five CMSs and that some genes were CMS-specific. Furthermore, essential genes in the cholesterol biosynthetic pathway can be identified if a cholesterol uptake reaction does not occur in the medium used. By contrast, the genes in the cholesterol biosynthetic pathway were determined to be nonessential if a cholesterol uptake reaction occurred in the medium used.

## Supporting information

**S1 Table. Uptake reactions in DMEM, Ham's medium and VMH medium.**
(XLSX)

**S2 Table. Cell mortality grades and metabolic deviation grades for essential genes for each CMS, obtained using the DMEM medium and various weighting factors.**
(XLSX)

**S3 Table. Cell mortality grades and metabolic deviation grades of essential genes for each consensus molecular subtype, obtained using various media.**
(XLSX)

**S4 Table. Cell mortality grades and metabolic deviation grades of combination of essential genes for each consensus molecular subtype, obtained using various media.**
(XLSX)

**S5 Table. Cell mortality grades and metabolic deviation grades of essential metabolites for each consensus molecular subtype, obtained using various media.**
(XLSX)

**S6 Table. Cell mortality grades and metabolic deviation grades of essential reactions for each consensus molecular subtype, obtained using various media.**
(XLSX)

**S1 Text. Nested hybrid differential evolution algorithm.** The source programs for identifying essential targets are coded using the General Algebraic Modeling System.
(PDF)

## Author Contributions

**Conceptualization:** Jin-Mei Lai, Peter Mu-Hsin Chang, Yi-Ren Hong, Chi-Ying F. Huang, Feng-Sheng Wang.

**Data curation:** Chao-Ting Cheng, Feng-Sheng Wang.

**Formal analysis:** Chi-Ying F. Huang, Feng-Sheng Wang.

**Funding acquisition:** Jin-Mei Lai, Peter Mu-Hsin Chang, Yi-Ren Hong, Chi-Ying F. Huang, Feng-Sheng Wang.

**Investigation:** Chao-Ting Cheng, Jin-Mei Lai, Peter Mu-Hsin Chang, Yi-Ren Hong, Chi-Ying F. Huang, Feng-Sheng Wang.

**Methodology:** Chao-Ting Cheng, Feng-Sheng Wang.

**Project administration:** Feng-Sheng Wang.

**Resources:** Chao-Ting Cheng, Feng-Sheng Wang.

**Software:** Chao-Ting Cheng, Feng-Sheng Wang.

**Supervision:** Jin-Mei Lai, Peter Mu-Hsin Chang, Yi-Ren Hong, Chi-Ying F. Huang, Feng-Sheng Wang.

**Validation:** Jin-Mei Lai, Peter Mu-Hsin Chang, Yi-Ren Hong, Chi-Ying F. Huang, Feng-Sheng Wang.

**Visualization:** Chao-Ting Cheng, Feng-Sheng Wang.

**Writing – original draft:** Jin-Mei Lai, Peter Mu-Hsin Chang, Yi-Ren Hong, Chi-Ying F. Huang, Feng-Sheng Wang.

**Writing – review & editing:** Jin-Mei Lai, Peter Mu-Hsin Chang, Yi-Ren Hong, Chi-Ying F. Huang, Feng-Sheng Wang.

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
