## [Decision Letter · Decision Letter 0]

3 Apr 2023

PONE-D-23-05789Identifying Essential Genes in Genome-Scale Metabolic Models of Consensus Molecular Subtypes of Colorectal CancerPLOS ONE

Dear Dr. Wang,

Thank you for submitting your manuscript to PLOS ONE. After careful consideration, we feel that it has merit but does not fully meet PLOS ONE’s publication criteria as it currently stands. Therefore, we invite you to submit a revised version of the manuscript that addresses the points raised during the review process.

Please submit your revised manuscript by **14th May, 2023.** If you will need more time than this to complete your revisions, please reply to this message or contact the journal office at plosone@plos.org. Please include the following items when submitting your revised manuscript:A rebuttal letter that responds to each point raised by the academic editor and reviewer(s). You should upload this letter as a separate file labeled 'Response to Reviewers'.A marked-up copy of your manuscript that highlights changes made to the original version. You should upload this as a separate file labeled 'Revised Manuscript with Track Changes'.An unmarked version of your revised paper without tracked changes. You should upload this as a separate file labeled 'Manuscript'.If applicable, we recommend that you deposit your laboratory protocols in protocols.io to enhance the reproducibility of your results. Protocols.io assigns your protocol its own identifier (DOI) so that it can be cited independently in the future. For instructions see: https://journals.plos.org/plosone/s/submission-guidelines#loc-laboratory-protocols. Additionally, PLOS ONE offers an option for publishing peer-reviewed Lab Protocol articles, which describe protocols hosted on protocols.io. Read more information on sharing protocols at https://plos.org/protocols?utm_medium=editorial-email&utm_source=authorletters&utm_campaign=protocols.

We look forward to receiving your revised manuscript.

Kind regards,

Bashir Sajo Mienda, PhD

Academic Editor

PLOS ONE

Journal Requirements:

Reviewers' comments:

Reviewer's Responses to Questions

**Comments to the Author**

1. Is the manuscript technically sound, and do the data support the conclusions?

Reviewer #1: Yes

2. Has the statistical analysis been performed appropriately and rigorously? 

Reviewer #1: No

3. Have the authors made all data underlying the findings in their manuscript fully available?

Reviewer #1: Yes

4. Is the manuscript presented in an intelligible fashion and written in standard English?

Reviewer #1: Yes

5. Review Comments to the Author

Reviewer #1: The authors investigate acquired metabolic sensitivity in different types of colorectal cancer. They use a published definition to assign samples retrieved from the TCGA database into 4 specific and 1 unknown molecular subtype, based on gene expression. They then use this data to reconstruct 6 genome scale metabolic models (5 disease and 1 healthy) and to score their metabolic reactions into 4 confidence levels. They set up a complex hierarchical multi-objective optimization problem to simultaneously estimate the flux in the cells and their acquired sensitivity to knock outs of reactions. This involves solving a flux balance analysis (FBA) problem while minimizing the L2 norm of fluxes weighted by the calculated confidence. The effects of simulated knockouts are graded by their effects on 3 metrics corresponding to 1) the difference in flux between healthy and diseased; 2) the effect on growth and ATP synthesis in the cancer model; 3) the same for healthy model. While the work seems technically sound, it is not clear if its assumptions can be biologically justified and thereby if it has relevance for addressing sensitivity in cancer. Its advantages over current approaches are not quantified.

Major comments

The approach relies on gene expression data to decide on reaction inclusion. While this is commonly done, it should be noted that genes must be translated to proteins for reactions to occur and differences in protein stability and other effects may influence protein levels so that mRNA levels do not accurately reflect them. Furthermore, an expression level below the limit of detection does not necessarily imply that a protein is inactive, since catalytic efficiency varies by orders of magnitude. For tumor samples, genes expression levels may be strongly influenced by the presence of infiltrating immune cells, so observed differences between healthy- and diseased samples may not necessarily reflect changes in the colon cells. It may be appropriate to discuss these limitations in the data used.

Genome scale metabolic models are underdetermined and there is a large range of flux distributions that may satisfy an objective function. The predicted fluxes are therefore dependent on the modeling choices. Often minimization of the L1 norm (parsimonious FBA) is used to select a unique solution. In this work a weighted L2 norm is used. It is based on the 4 confidence levels derived from gene expression data and they are weighted with factors (0.25, 0.5, 0.75, 1). This weighing is a modeling decision, and it could be of interest to show how the results would be affected by different choices. In particular since the difference between healthy and diseased fluxes, is used in the evaluation and may be affected. One alternative to quantized confidence scores would be a continuous value.

The objective function for healthy cells is set to maximize ATP synthesis. It is unclear if this is a realistic assumption about what constitutes a healthy state. Furthermore, in absence of limitations on uptake reactions and other constraints, FBA will increase the flux through the objective function until it reaches one of the artificial bounds on internal fluxes (often -1000 or 1000). Is this occurring in these simulations or is this prevented somehow. TAre the predicted fluxes reasonable for human cells? Fluxes of around 13.649 mmol/gDW h, are reported, is this level in line with published values for healthy colon cancer cells? It could also be interesting to comment on the difference between culture medium and the in vivo microenvironment, where perhaps metabolites, such as cholesterol, may be present.

The abstract discusses that gene essentiality prediction is a time-consuming process. However, a brute force method is used in this study to validate the essentiality predictions. It would be useful to quantify the speed up that can be achieved with the proposed approach.

In table 4 the cell mortality grade is reported. It appears to be strikingly similar for all genes within each molecular subtype. Is there an explanation to this, e.g. how the objective function is set up. For the metabolic deviation grade metric, the fluxes of the cancer cell model is compared with the healthy flux distribution. Is it always the same flux distribution for the healthy or does it vary between models? It is stated that one objective is to maximize the cell viability of perturbed healthy cells during gene knockout or deletion. Why is it assumed that healthy cells will be resistant to perturbations?

The model makes predictions about gene essentiality and the predictions are compared with observations in DepMap. It would be useful to quantify the prediction performance, e.g. in the form of a confusion matrix (with true and false positives) and to apply an appropriate statistical tests. It could also be interesting to speculate on what may be the reason that some predicted essential genes were not essential in DepMap.

What was the rational for perturbing essential genes together with non-essential?

Minor comments

It is hard to tell from the methods if any work was done in this study to classify the TCGA samples. If the classification from Guinney 2015 was used without modification, then the corresponding methods section could be simplified.

Table 3 does not appear to be showing what was intended, since the first 2 lines for each species show the exact same value, even though there seems to be multiple unique entities for each model.

It implied that cells with 0 growth rate are dead, however cells can also enter a state of senescence where they neither grow nor die.

It is mentioned that “inhibition of formate production reduced intracellular cholesterol synthesis and thereby eliminated the growth of cancer cells”. Why is growth eliminated by a mere reduction in biosynthesis rate?

The use of reaction identifiers specific to the recon model, such as ‘R_OMPDC’ may perhaps not be very informative to most readers.

Perhaps the sentence "Formate is an intermediate metabolite in one-carbon metabolism, and acts as the source and sink in mammalian cell metabolism" may need further clarification about what is considered source and sink.

6. PLOS authors have the option to publish the peer review history of their article (what does this mean?). If published, this will include your full peer review and any attached files.

Reviewer #1: No

---

## [Author Response · Author response to Decision Letter 0]

18 Apr 2023

We have discussed the influence effects of the weighting factors assigned and added two tables (S2 Table) in the supplementary in the revised manuscript, and drawn Table 3 to a bar chart as Figure 3 in the revised manuscript to clarify the meaning of the data for the models. We have explained the reviewer comments in the attached file.

---

## [Decision Letter · Decision Letter 1]

7 May 2023

Identifying Essential Genes in Genome-Scale Metabolic Models of Consensus Molecular Subtypes of Colorectal Cancer

PONE-D-23-05789R1

Dear Dr. wang,

We’re pleased to inform you that your manuscript has been judged scientifically suitable for publication and will be formally accepted for publication once it meets all outstanding technical requirements.

Kind regards,

Bashir Sajo Mienda, PhD

Academic Editor

PLOS ONE

Additional Editor Comments (optional):

Reviewers' comments:

Reviewer's Responses to Questions

**Comments to the Author**

1. If the authors have adequately addressed your comments raised in a previous round of review and you feel that this manuscript is now acceptable for publication, you may indicate that here to bypass the “Comments to the Author” section, enter your conflict of interest statement in the “Confidential to Editor” section, and submit your "Accept" recommendation.

Reviewer #1: All comments have been addressed

2. Is the manuscript technically sound, and do the data support the conclusions?

Reviewer #1: Yes

3. Has the statistical analysis been performed appropriately and rigorously? 

Reviewer #1: No

4. Have the authors made all data underlying the findings in their manuscript fully available?

Reviewer #1: Yes

5. Is the manuscript presented in an intelligible fashion and written in standard English?

Reviewer #1: No

6. Review Comments to the Author

Reviewer #1: The authors have addressed the comments. It should perhaps be noted that while the language of the manuscript is generally clear, the revised sections would benefit from proofreading by a native speaker.

7. PLOS authors have the option to publish the peer review history of their article (what does this mean?). If published, this will include your full peer review and any attached files.

Reviewer #1: No

---

## [Editor Report · Acceptance letter]

11 May 2023

PONE-D-23-05789R1 

Identifying Essential Genes in Genome-Scale Metabolic Models of Consensus Molecular Subtypes of Colorectal Cancer 

Dear Dr. Wang:

I'm pleased to inform you that your manuscript has been deemed suitable for publication in PLOS ONE. Congratulations! Your manuscript is now with our production department. 

Kind regards, 

on behalf of

Dr. Bashir Sajo Mienda 

Academic Editor

PLOS ONE